# Dose-Dependent Impairment of the Immune Response to the Moderna-1273 mRNA Vaccine by Mycophenolate Mofetil in Patients with Rheumatic and Autoimmune Liver Diseases

**DOI:** 10.3390/vaccines10050801

**Published:** 2022-05-18

**Authors:** Maria De Santis, Francesca Motta, Natasa Isailovic, Massimo Clementi, Elena Criscuolo, Nicola Clementi, Antonio Tonutti, Stefano Rodolfi, Elisa Barone, Francesca Colapietro, Angela Ceribelli, Matteo Vecellio, Nicoletta Luciano, Giacomo Guidelli, Marta Caprioli, Clara Rezk, Lorenzo Canziani, Elena Azzolini, Luca Germagnoli, Nicasio Mancini, Ana Lleo, Carlo Selmi

**Affiliations:** 1Division of Rheumatology and Clinical Immunology, IRCCS Humanitas Research Hospital, 20089 Rozzano, Milan, Italy; maria.de_santis@hunimed.eu (M.D.S.); francesca.motta2@humanitas.it (F.M.); natasa.isailovic@humanitasresearch.it (N.I.); angela.ceribelli@hunimed.eu (A.C.); matteo.vecellio@humanitasresearch.it (M.V.); nicoletta.luciano@humanitas.it (N.L.); giacomo.guidelli@humanitas.it (G.G.); marta.caprioli@humanitas.it (M.C.); lorenzo.canziani@humanitas.it (L.C.); 2Department of Biomedical Sciences, Humanitas University, 20090 Pieve Emanuele, Milan, Italy; antonio.tonutti@humanitas.it (A.T.); stefano.rodolfi@humanitas.it (S.R.); elisa.barone@hunimed.eu (E.B.); calara.rezk@st.hunimed.eu (C.R.); ana.lleo@hunimed.eu (A.L.); 3Laboratory of Medical Microbiology and Virology, University Vita-Salute San Raffaele, 20132 Milan, Italy; clementi.massimo@hsr.it (M.C.); clementi.nicola@hsr.it (N.C.); mancini.nicasio@hsr.it (N.M.); 4IRCCS San Raffaele Hospital, 20132 Milan, Italy; criscuolo.elena@hsr.it; 5IRCCS Humanitas Research Hospital, 20089 Rozzano, Milan, Italy; 6Division of Internal Medicine and Liver Disease, IRCCS Humanitas Research Hospital, 20089 Rozzano, Milan, Italy; francesca.colapietro@humanitas.it; 7Medical Direction, IRCCS Humanitas Research Hospital, 20089 Rozzano, Milan, Italy; elena.azzolini@humanitas.it; 8Diagnostic Laboratory, IRCCS Humanitas Research Hospital, 20089 Rozzano, Milan, Italy; luca.germagnoli@humanitas.it

**Keywords:** COVID-19, vaccination, mycophenolate mofetil, autoimmune diseases, antirheumatic agents

## Abstract

The purpose of this study was to evaluate the efficacy and safety of the Moderna-1273 mRNA vaccine for SARS-CoV-2 in patients with immune-mediated diseases under different treatments. Anti-trimeric spike protein antibodies were tested in 287 patients with rheumatic or autoimmune diseases (10% receiving mycophenolate mofetil, 15% low-dose glucocorticoids, 21% methotrexate, and 58% biologic/targeted synthetic drugs) at baseline and in 219 (76%) 4 weeks after the second Moderna-1273 mRNA vaccine dose. Family members or caretakers were enrolled as the controls. The neutralizing serum activity against SARS-CoV-2-G614, alpha, and beta variants in vitro and the cytotoxic T cell response to SARS-CoV-2 peptides were determined in a subgroup of patients and controls. Anti-SARS-CoV-2 antibody development, i.e., seroconversion, was observed in 69% of the mycophenolate-treated patients compared to 100% of both the patients taking other treatments and the controls (*p* < 0.0001). A dose-dependent impairment of the humoral response was observed in the mycophenolate-treated patients. A daily dose of >1 g at vaccination was a significant risk factor for non-seroconversion (ROC AUC 0.89, 95% CI 0.80–98, *p* < 0.0001). Moreover, in the seroconverted patients, a daily dose of >1 g of mycophenolate was associated with significantly lower anti-SARS-CoV-2 antibody titers, showing slightly reduced neutralizing serum activity but a comparable cytotoxic response compared to other immunosuppressants. In non-seroconverted patients treated with mycophenolate at a daily dose of >1 g, the cytotoxic activity elicited by viral peptides was also impaired. Mycophenolate treatment affects the Moderna-1273 mRNA vaccine immunogenicity in a dose-dependent manner, independent of rheumatological disease.

## 1. Introduction

Since the approval of anti-SARS-CoV-2 vaccines at the end of 2020, there have been questions about their efficacy and safety in patients with immune-mediated and chronic inflammatory diseases taking immunosuppressive therapies as randomized clinical trials were performed largely on healthy individuals [1,2]. Furthermore, only a few studies analyzed the antibody-neutralizing activity against different viral variants [3,4,5,6,7,8] or the cellular response [9,10,11] following vaccination in these fragile patients. 

It has been suggested that specific immunomodulatory drugs may reduce the immune response to vaccines [12,13,14]. However, data regarding the mechanisms or magnitude of this phenomena are limited. Considering that these drugs modulate the immune system, and seeing as the mRNA vaccines generate protective immunity by eliciting SARS-CoV-2-specific CD4+ and CD8+ T cells along with neutralizing antibodies [12], it may be hypothesized that immunosuppressive therapies affecting T cell function or antibody production are expected to impact the immune response to the vaccine. 

As an example, glucocorticoids are broad spectrum immunosuppressants and not only reduce the number of circulating B cells and, respectively, IgAs and IgGs, but also impair phagocyte and T cell function and lead to T cell apoptosis [13]. Methotrexate reduces antigen-dependent T cell proliferation [14]. Mycophenolate mofetil, metabolized into mycophenolic acid, is a selective inhibitor of inosine monophosphate dehydrogenase type II, and is involved in the first step for the formation of guanine ribonucleotides [15]. This enzyme is strongly expressed in activated B cells, thus explaining the impact of this drug in diseases where B cells play a key pathogenic role, such as systemic lupus erythematosus or systemic sclerosis, or in vaccine humoral response. Moreover, mycophenolate impairs dendritic cell maturation [16] and, consequently, the immune response to both external and self-antigens. The drug acid also exerts an anti-proliferative effect on T cells [17], which was specifically studied for the influenza virus antigen-specific CD8+ T cells. It has been demonstrated that treatment with mycophenolate does not lead to a significant inhibition of cytotoxic activity mirrored by granzyme B production, while reducing CD8+ T cell proliferation and memory phenotype development. On the contrary, mycophenolate also significantly inhibits cytotoxic activity [15], thus posing the question on the need for mycophenolate withdrawal in patients to achieve an adequate vaccine immune response, bearing in mind the feasibility of halting therapy in patients at higher risk of disease flare.

Translating the observed changes into clinical recommendations, glucocorticoids have been shown to impair both vaccine and cellular responses to tetanus and diphtheria vaccines; similarly, methotrexate has been associated with a poor vaccine response, especially with conjugated polysaccharide pneumococcal and protein antigen influenza vaccines [18,19,20]. Other studies have shown that the temporal withholding of methotrexate after vaccination improved the immunogenicity of the seasonal influenza vaccination in patients affected by rheumatoid arthritis [21]. Based on the available data, the American College of Rheumatology (ACR) proposed a guidance in March 2021 [22] that suggested a short-term withdrawal of methotrexate, JAK inhibitors, and abatacept and a deferral of rituximab and cyclophosphamide infusions, when clinically feasible, to maximize the immune response. A later update in August 2021 suggested the additional withdrawal of mycophenolate [23]. 

With the progressive availability of the COVID-19 vaccine and the subsequent studies performed on fragile patients, earlier data reported lower seroconversion rates and anti-SARS-CoV-2 immunoglobulin G (IgG) titers in patients receiving abatacept, rituximab, and mycophenolate mofetil [24,25,26,27,28,29,30]. Correspondingly, a multicenter observational study evaluating the immunogenicity and safety of the BNT162b2 mRNA vaccine in 686 adult rheumatological patients compared to the general population showed that patients treated with glucocorticoids, mycophenolate mofetil, abatacept, JAK-inhibitors, and rituximab exhibited reduced immunogenicity to the vaccine [29]. Likewise, another prospective study found that JAK-inhibitors and methotrexate both reduced antibody titers, while other biologic therapies had limited effects on titer levels [31]. Although data on the isolated effect of mycophenolate treatment on COVID-19 vaccine immunogenicity is limited, a recent study assessing the antibody titer and seroconversion rates after a two-dose mRNA vaccine or a single-dose adenoviral vector vaccine in mycophenolate-treated systemic sclerosis patients showed that the use of mycophenolate was associated with a negative antibody response. More specifically, patients treated with mycophenolate at doses higher than >2.5 g daily were more likely to have a negative antibody response, and those with a dose higher than 3 g daily were less likely to have an antibody response at all [32]. In a similar vein, another multicenter prospective observational study on the immune response to SARS-CoV-2 vaccination in kidney transplant patients showed that of the patients treated with mycophenolate, those assuming a dose of >1 g per day did not achieve seroconversion or produce neutralizing antibodies [33]. 

We herein investigated the efficacy, in terms of humoral and cellular response, and safety of two doses of the Moderna-1273 mRNA vaccine in a cohort of patients with rheumatic and autoimmune liver diseases.

## 2. Materials and Methods

### 2.1. Participants and Setting

Adult patients (*n* = 1073) followed at the Rheumatology and Clinical Immunology and Liver Disease Outpatient Clinics of the Humanitas Research Hospital were directly solicited by phone to undergo the anti-SARS-CoV-2 vaccination using the mRNA-1273 vaccine (Moderna) in March 2021. Out of these, 287 patients agreed to participate; the main reasons why the other solicited patients did not participate in the study were two-fold. First, at the time of the study, Italian citizens with chronic diseases were likely to be solicited for vaccination by different agencies/health care providers for other factors such as older age or professional risks. Second, we should note that more than 30% of patients followed at our Center came from a different region and preferred to get vaccinated in their geographical area. All the patients participating in the study provided written consent (following the protocol approval by the Humanitas ethical committee) and received two doses of vaccine 4 weeks apart (17–19 April and 19–22 May 2021). Sixty-seven caregivers or unrelated family members of the patients were included as the controls if they were not affected by immune-mediated or oncological disease. Enrolled patients had no history of cancer within the preceding 5 years and were on a stable dose of medical therapy for at least 4 weeks prior to vaccination. JAK inhibitors, methotrexate, abatacept, and rituximab were discontinued while other medications were unchanged, in accordance with the American College of Rheumatology recommendations available as of April 2021 [22]. A previous history of COVID-19 and/or positive serology at baseline defined the ex-COVID status, while patients without previous COVID-19 and with undetectable antiSARS-CoV-2 IgG at baseline were defined as COVID-naïve.

### 2.2. Study Design

In this observational prospective cohort study, we obtained peripheral blood samples from 287 patients and 67 controls on the day of the first vaccine dose (T0), 281 (98%) patients and 65 (97%) controls on the day of the second dose 4 weeks later (T1), and 219 (76%) patients and 45 (67%) controls 4 weeks after the second dose (T2). The dropouts were due mainly to the refusal of some subjects to undergo a new blood withdrawal. To evaluate the late impact on immune response, we added a further timepoint 8 weeks after the second dose (T3), where we analyzed only the patients, 47 in total; in particular, those under mycophenolate (10 patients agreed to continue the study) compared with other 37 patients (22 treated with anti-TNF-alpha, 6 with JAK inhibitors, and 9 not on immunosuppressive treatment). This latter population was randomly selected and constituted a representative sample of the initial population.

Clinical data, including drug dosages and disease activity, were obtained from the medical records a maximum of 8 weeks before T0. All patients and controls were asked about previous COVID-19 infection at T0 and about side effects at T1 and T2. Serum samples were tested for anti-SARS-CoV-2 antibodies. A subgroup of patients representing different conditions and treatments were tested for the neutralizing activity against the wild-type virus and two viral strains: the B.1.1.7 (alpha or UK) and B.1.351 (beta or South African) variants. These variants and their peripheral blood mononuclear cells (PBMCs) were used for cytotoxic assay. Sixteen weeks after the second dose, we performed a phone interview of 276/287 patients and 67 controls to inquire about disease activity, such as the worsening of pain, skin signs and symptoms, fatigue, dyspnea, or blood test modifications (especially regarding liver enzymes and inflammatory markers) and about incident COVID-19 infections that occurred after the first and the second dose of vaccine.

### 2.3. Anti-SARS-CoV-2 Antibodies

IgG against the subunits S1 and S2 of the spike protein were quantitatively measured using an indirect chemiluminescent immunoassay (CLIA) (LIAISON^®^, DiaSorin, Saluggia, Italy). Seroconversion was defined as values of ≥ 15 Arbitrary Unit (AU)/mL. We also quantitatively detected IgG directed towards a new generation recombinant SARS-CoV-2 trimeric spike glycoprotein using an indirect CLIA (LIAISON^®^, DiaSorin, Saluggia, Italy). Seroconversion was defined as values ≥ 33.8 Binding Antibody Unit (BAU)/mL. 

### 2.4. Serum Microneutralization Assay

We investigated the capacity to neutralize SARS-CoV-2 variants of serum obtained at T2 from a selection of 17 patients and 5 controls representing immunosuppressive therapy groups, previous COVID-19 history, and seroconversion status.

Vero E6 (Vero C1008, clone E6—CRL-1586; ATCC) cells were cultured in Dulbecco’s modified Eagle’s medium (DMEM) supplemented with non-essential amino acids (NEAA), penicillin/streptomycin (P/S), Hepes buffer, and 10% (*v*/*v*) fetal bovine serum (FBS). Three clinical isolates of SARS-CoV-2 were obtained and propagated in Vero E6 cells: G614 (hCoV-19/Italy/UniSR1/2020; GSAID Accession ID: EPI_ISL_413489), B.1.1.7 (alpha or “UK” variant-19/Italy/LOM-UniSR7/2021; and GSAID Accession ID: EPI_ISL_1924880), B.1.351 (beta or “South African” variant-hCoV-19/Italy/LOM-UniSR6/2021, GISAID Accession ID: EPI_ISL_1599180). Vero E6 cells were seeded into 96-well plates 24 h before the experiment at 95% cell confluence for each well. Serum samples were decomplemented at 56 °C for 30 min and incubated at a 1:80 dilution with SARS-CoV-2 variants at 0.01 multiplicity of infection (MOI) for 1 h at 37 °C. Virus–serum mixtures and a positive infection control were applied to Vero E6 monolayers after washing cells with PBS 1×, and virus adsorption was carried out at 37 °C for 1 h. Then, the cells were washed with PBS 1× to remove cell-free virus particles and complete DMEM supplemented with 2% FBS, which was added to the cells. The plates were incubated at 37 °C in the presence of CO2 for 72 h. The experiments were performed in triplicate. Neutralization activity was evaluated by comparing the cytopathic effect detected in the presence of virus–serum mixtures to the positive infection control.

### 2.5. Cytotoxic T Cell Response and Chemokine/Cytokine Levels

PBMCs from 14 patients (representing groups of immunosuppressive therapies) and 4 controls at T2 were isolated using Vacutainer CPT Mononuclear Cell Preparation Tube-Sodium Heparin, IVD, 8 mL according to manufacturer’s protocol and 2 × 10^5^ were seeded in duplicate for every condition in a 96-well U-bottom plate in complete RPMI1640 medium. After 24 h in a resting state, the cells were stimulated (or not) with 200 ng/mL of SEB (Merck KGaA, Darmstadt, Germany) or with 700 ng/mL with peptivator (Miltenyi, Bergish Gladbach, Germany) for another 24 h at 37 °C in the presence of CO2. After incubation, the supernatants were collected and the cytokine levels were measured using a MACSPlex cytotoxic T/NK cell kit (Miltenyi, Bergish Gladbach, Germany) based on three bead populations that had been coated with capture antibodies specific to the following proteins: granzyme B, IFN-gamma, perforin, GM-CSF, IL-2, IL-21, MCP-1, IL-10, IL-6, IL-17, and TNF-alpha, and distinguished by channel FITC and PE by a flow cytometer. These capture beads were added to the supernatant samples of unknown concentrations and to a serially diluted MACSPlex cytotoxic cell standard, following incubations with detection reagents containing APC. The beads were acquired using a flow cytometer (Fortessa BD). A standard curve was generated. The median of the APC fluorescence of each capture bead population and concentration was calculated using the FlowLogic program (Datanova, Australia). The values were normalized by subtracting levels in unstimulated cultures to the levels of the peptivator-stimulated culture. The SEB culture results were used to test the normal cell immune response of the cells to the bacterial products.

### 2.6. Statistical Analysis 

The results were described as numbers (percentages), mean plus standard deviation (SD) or standard error (SEM), or median and interquartile range (IQR). The Mann–Whitney test, Wilcoxon test, or Receiving Operating Characteristic (ROC) curve were used, as appropriate, and K statistics was used to calculate the agreement between the two assays used to detect anti-SARS-CoV-2 antibodies (GraphPAD 7.02). When analyzing the humoral response according to the immunosuppressive treatments, only COVID-19-naïve patients were included to avoid biases related to previous infection. The comparison between groups was not feasible for neutralizing antibody titers and cytotoxicity assays due to the small number of observations in each group.

## 3. Results

### 3.1. Study Subjects

The general features of the enrolled patients and controls are illustrated in Table 1. 

The majority of the patients were affected by inflammatory arthritis (69%), followed by autoimmune liver disease (16%) and connective tissue disease (10%). At enrollment, 28% of the patients had an active disease (70% with arthritis and 15% with connective tissue diseases). Twenty-eight patients (10%) were taking mycophenolate (with a median dose of 1750, IQR 1000–2875 mg/day), the majority of which had systemic sclerosis. Fifteen % of patients were taking low-dose systemic glucocorticoids (prednisone-equivalent daily median dose of 5, interquartile range (IQR) of 3.75–5 mg/day) and 40% were taking conventional synthetic disease-modifying anti-rheumatic drugs (DMARDs), most frequently, methotrexate (21%, with a median dose of 10, IQR 10–15 mg/week) or azathioprine (9%, with a median dose of 100, i.e., 1–2 mg/kg, IQR 50–100 mg/day). Biologic/targeted synthetic DMARDs were used by 58% of patients, mainly anti-TNFalpha (38% total, subdivided as 43% adalimumab, 30% etanercept, 15% golimumab, 9% certolizumab, and 3% infliximab). Eighteen (6%) patients were not taking immunosuppressive drug therapies at the time of vaccination, and this included 13 patients with primary biliary cholangitis that were taking ursodeoxycholic acid. Of the 287 patients and 67 controls, 45 (16%) and 15 (22%) were found to be ex-COVID, respectively (*p* > 0.05; Table 1).

### 3.2. Anti-SARS-CoV-2 Antibody Response in the Overall Cohort 

The two antibody tests (IgG against the subunits S1 and S2 or IgG against a recombinant SARS-CoV-2 trimeric spike glycoprotein) had an agreement of 97% (expected agreement 54%, Kappa 0.9332, SEM 0.0322, Z 28.99, *p* < 0.0001). We illustrate the results throughout the manuscript, referring to the anti-trimeric spike glycoprotein test. At T1, anti-SARS-CoV-2 antibodies were positive in 100% of the ex-COVID patients and controls, as expected, while seroconversion was observed in 82% of the COVID-naïve patients and 98% of the COVID-naïve controls (COVID-naïve patients vs. COVID-naïve controls *p* = 0.008, OR 10.6, 95% CI 1.7–79). Moreover, the COVID-naïve patients had significantly lower antibody titers compared to the COVID-naïve controls (*p* < 0.0001), while in the ex-COVID group, no significant difference in antibody titers was detected between the patients and controls (Table 2 and Figure 1). At T2, anti-SARS-CoV-2 antibodies were positive in 96% of the patients and 100% of the controls. Similar to T1, the COVID-naïve patients had significantly lower antibody titers compared to the COVID-naïve controls (*p* = 0.0002), while the ex-COVID patients did not have significantly different antibody titers compared to the ex-COVID controls (*p* = 0.34; Table 2 and Figure 1). The patients not receiving immunosuppressive drugs at the time of vaccination had similar anti-SARS-CoV-2 titers compared to the controls (Table 2 and Figure 1). The antibody titers at T2 were inversely correlated with age among the controls (*p* = 0.01, r = −0.39), but not among the patients (*p* = 0.050, r = −0.14). The disease activity (Table 2) and disease duration did not influence anti-SARS-CoV-2 antibody seroconversion rates or antibody titers. 

### 3.3. Seroconversion Rate Based on Active Treatments

Eight patients (8/219, 4%) did not achieve seroconversion at T2 and were all on mycophenolate (8/26, 31% of mycophenolate-treated patients at T2; *p* < 0.0001 vs. other drugs); moreover, they were all COVID-naïve patients affected by systemic sclerosis (8/12, 67% of patients with systemic sclerosis; *p* < 0.0001 vs. other patients). The daily dosage of mycophenolate was significantly higher in non-serocoverted patients (median dose of 2250 (IQR 1625–3000) mg/day) compared to seroconverted mycophenolate-treated patients (1250 (IQR 1000–2000) mg/day; *p* = 0.02). The seroconversion rate at T2 in mycophenolate-treated patients was 69% (18/26), with a lower rate in patients treated with a dosage higher than 1 g daily (60%, 12/20). An ROC curve identified a mycophenolate daily dose of >1 g as a significant risk factor for lack of seroconversion at T2 (AUC 0.89, sensitivity 73%, specificity 100%, *p* < 0.0001, 95% CI 0.80–98). Age, sex, disease activity, and duration did not affect the seroconversion rate. Details on seroconversion rates and ongoing therapies in COVID-naïve patients are shown in Table 3.

### 3.4. Anti-SARS-CoV-2 Antibody Titer Based on Ongoing Therapies

At T2, anti-SARS-CoV-2 titers were significantly lower compared to the controls only in the patients taking mycophenolate (*p* < 0.0001, Table 3); moreover, the patients on mycophenolate had significantly lower anti-SARS-CoV-2 titers than the patients treated with any other immunosuppressants (Table 3). The mycophenolate dosage significantly impacted IgG titers, as patients treated with >1 g/daily had lower titers compared to patients taking ≤1 gr/daily (43.2 (16–443.5) vs. 1280 (212–6458), respectively; *p* = 0.004). Details on antiSARS-Cov-2 titers and ongoing therapies in the COVID- naïve patients are shown in Table 3, while those regarding patients on mycophenolate >1 g/daily are shown in Table 4. 

For this latter category, the total lymphocyte count at baseline (considered as a marker of potential adaptive immunity function) was within normal range in most patients. A subgroup of 10 patients treated with mycophenolate at the time of vaccination (5 of which had not developed an antibody response at 4 weeks) were tested at T3 (data not shown in tables). AntiSARS-CoV-2 antibodies were positive in 3/5 previously seronegative patients, although at low titers (with a range of 41.4–136 BAU/mL). Cumulatively, in the patients treated with mycophenolate, the IgG titers at T3 (88.2 (36.2–569.5) BAU/mL) were higher compared to T2 (34.2 (12.5–193) BAU/mL), although not reaching a statistically significant difference (*p* = 0.25). On the other hand, in the other 37 patients (22 treated with anti-TNF-alpha, 6 with JAK inhibitors, and 9 not on immunosuppressive treatment), the T3 anti-SARS-CoV-2 titers were significantly lower compared to those at T2 (1490 (873.5–2056) at T3 vs. 4020 (2055–6128) at T2 BAU/mL, *p* < 0.0001). These data suggest a delayed anti-SARS-CoV-2 antibody peak in the patients treated with mycophenolate.

### 3.5. Anti-SARS-CoV-2 Antibody-Neutralizing Activity in a Subgroup of Patients and Controls

The in vitro neutralizing activity against the SARS-CoV-2 G614, UK, and South African variants was tested on the sera from 17 patients (5 ex-COVID) and 5 controls (3 ex-COVID). The ex-COVID patients included two treated with mycophenolate at 3 g/day, two with baricitinib at 4 mg/day, and one with methotrexate at 15 mg/week. The 12 COVID-naïve patients included 2 treated with mycophenolate at 3 g/day, 2 with baricitinib at 4 mg/day, 2 with methotrexate at 15 mg/week, 2 with secukinumab at 150 mg/4 weeks, and 4 with adalimumab at 40 mg/2 weeks. Statistical analysis was not performed considering the small sample size; nonetheless, the neutralizing activity seemed similar in the ex-COVID patients compared to controls, while it was reduced in the COVID-naïve patients (Figure 2, panel A). When separately considering the mycophenolate-treated patients, a reduced neutralizing activity was observed compared to the other patients (Figure 2, panel B). 

### 3.6. Cytotoxic T Cell Response Markers in a Subgroup of Patients and Controls

The cytotoxic T cell response after stimulation with SARS-CoV-2 mixed peptides mirrored by perforin, IFN-gamma, and granzyme B levels was also studied. Statistical analysis was not performed considering the small sample size; nonetheless, the cytotoxic T cell response did not seem different in the eight COVID-naïve patients treated with different immunosuppressants (ongoing at time T2 of blood sample collection: two treated with mycophenolate at 3 g/day, three with baricitinib at 4 mg/day, and three with adalimumab at 40 mg/2 weeks) compared to the four COVID-naïve controls (Figure 3, panel A). The patients treated with mycophenolate did not show a reduced cytotoxic response compared to the other patients (Figure 3, panel A). However, in analyzing the patients based on mycophenolate treatment, seroconversion status, and previous history of COVID-19, a reduced T cell response was observed in the non-seroconverted mycophenolate-treated patients (Figure 3, panel B). In particular, only one of the three non-seroconverted patients had a detectable cytotoxic response (Figure 3, panel B). A patient suffering from limited systemic sclerosis who was positive for the antibody anti-topoisomerase 1 and was not being treated with immunosuppressants had cytotoxic marker levels similar to the mycophenolate-treated patients, suggesting that the connective tissue disease might have a limited influence (Figure 3, panel B).

### 3.7. Chemokine and Cytokine Levels after SARS-CoV-2 Peptide-Stimulation In Vitro 

The mean levels of GM-CSF and IL-2, both involved in increasing vaccine response [34,35] and able to act as vaccine adjuvants, were lower in 8 patients treated with mycophenolate (with a mean of 10 (SEM 11.1) and 2837 (3063) pg/mL, respectively) or in 5 patients treated with baricitinib (13.6 (75.9) and 1408.6 (1654.6) pg/mL, respectively) compared to 4 patients treated with adalimumab (143.4 (47.2) and 157,384 (290,305.7) pg/mL, respectively) and to 3 healthy subjects (54.4 (77.2) and 2049.6 (1070) pg/mL, respectively). Of note, higher levels of MCP-1, a marker of vaccine-induced recruitment of suppressive myeloid cells that is able to inhibit vaccine-induced humoral immunity [36], were found in patients treated with mycophenolate (76,659 (158,142.6) pg/mL) and in patients treated with baricitinib (312,632.3 (197,658.2) pg/mL) compared to patients treated with adalimumab (zero) and to healthy subjects (zero in two subjects, 13,370 pg/mL in one subject). Taking these results together and considering that mycophenolate was not suspended at the time of the first and second dose, our data suggest that further mycophenolate-mediated mechanisms negatively impact the humoral response to the vaccine. Lower levels of inflammatory cytokines, such as IL-6 and IL-17, were found in the patients treated with mycophenolate (0 and 1.4 (9.18) pg/mL, respectively) and in patients treated with baricitinib (43.8 (64.7) and 0 pg/mL, respectively) compared to patients treated with adalimumab (1549 (93,535.9) and 99.6 (69.7) pg/mL, respectively) and to healthy subjects (102.3 (144.7) and 38.2 (63.2) pg/mL, respectively). Surprisingly, comparable levels of TNF-alpha were found in the patients treated with mycophenolate (120 (174.1) pg/mL), in patients treated with baricitinib (248 (539.1) pg/mL), and in healthy subjects (260.5 (299) pg/mL), while higher levels were found in patients treated with adalimumab (1707.8 (1931.1) pg/mL). We did not document different inflammatory cytokine levels in the subjects reporting adverse events after the second vaccine dose (data not shown). 

Similar levels of IL-21, correlating with B memory cell development [37], were found among all groups (mycophenolate 8.6 (7.6), baricitinib 9.2 (29.1), adalimumab 25.2 (11), and healthy subjects 9.8 (11.3) pg/mL). Lower levels of IL-10 were found in the patients treated with mycophenolate (113.7 (238.9) pg/mL) and in the patients treated with baricitinib (138.3 (128) pg/mL) compared to the patients treated with adalimumab (440.8 (393.4) pg/mL) and to the healthy subjects (310.5 (112) pg/mL).

### 3.8. Adverse Events and SARS-CoV-2 Breakthrough Infections 

The vaccine-related side effects were mild in most cases, with similar incidence rates in both patients and controls (Table 5), without differences based on ongoing therapies. Of note, the controls who reported adverse events developed higher antibody titers (*p* = 0.009). None of the 276 patients and 67 controls contacted 16 weeks following the second dose developed SARS-CoV-2 infection. Two patients (on anti-TNF-alpha) and one control had proven SARS-CoV-2 infection one week after the first vaccine dose; they all had mild symptoms and did not require hospitalization. 

## 4. Discussion

We report that the Moderna-1273 mRNA vaccine is effective and safe in patients with rheumatic or autoimmune liver diseases under immunosuppressive therapies, similar to what was observed with the 162b2 Pfizer/Biontech vaccine [29,38,39]. Our data suggest that whereas being affected by a specific autoimmune disease does not, per se, impair the response to vaccination, specific immunosuppressive agents affect immunogenicity, in particular, mycopenolate. In our cohort, the heterogeneous treatments with different mechanisms of action and impact on the immune system had influenced the overall results, with antibody titers being significantly lower in the patients under immunosuppressive therapy compared to the controls. In any case, our data allow us to draw some important considerations. Mycophenolate was the drug with the worst impact on the immune response, both in terms of seroconversion and antibody titer levels. On the other hand, the other immunosuppressants, both for their mechanisms of action and by adopting ACR guidance on short-term withdrawal for vaccination, had mild, if any, impacts on immunogenicity.

The patients discontinuing JAK inhibitors or methotrexate for 7 days after each dose had similar seroconversion rates and anti-SARS-CoV-2 IgG titers compared to the controls, supporting the validity of the ACR guidelines [23]. 

Mycophenolate mofetil (for which a withdrawal was not recommended at the time of this study [22]) significantly reduced both the rate of seroconversion in COVID-naïve patients and the anti-SARS-CoV-2 antibody titers in seroconverted patients, independent of the underlying disease. Moreover, mycophenolate also reduced the neutralizing activity of anti-SARS-CoV-2 antibodies against the wild-type virus and against the UK and SA variants compared to the other immunosuppressant-treated patients. 

Mycophenolate has already been described as an agent impairing the response to other vaccinations. A recent meta-analysis showed that mycophenolate lowered the seroconversion rate after influenza vaccination among transplant recipients with a relative risk of 0.7 (0.6–0.9) compared to other immunosuppressive agents [40]. In addition, in a large observational study on the BNT162b2 Pfizer anti-COVID19 vaccine conducted on 686 patients with rheumatic diseases, mycophenolate treatment predicted a poor vaccine response, with a seroconversion rate of 64% and an adjusted OR of 0.1 [29]. In our study, the mycophenolate-treated patients had a similar seroconversion rate, which was even lower in the patients treated with higher doses. Moreover, a delayed anti-SARS-CoV-2 IgG titer peak was observed, together with an altered expression of factors associated to successful vaccine immune response, such as lower GM-CSF and IL-2 levels and higher MCP-1 levels. In addition, in line with previous evidence [15], our data showed that the cytotoxic activity was not impaired by mycophenolate in the seroconverted patients. Conversely, only one out of three non-seroconverted patients treated with mycophenolate manifested a detectable cytotoxic activity. These data suggest that the achievement of seroconversion may reflect the development of a cytotoxic response during mycophenolate treatment. Of note, all the patients on mycophenolate of ≤1 g daily achieved seroconversion with higher IgG titers compared to patients taking >1 g daily, thus possibly maintaining a cytotoxic response to the vaccine. This finding continues to be observed in subsequent similar studies performed on diverse, fragile patient groups, including rheumatological patients and transplant recipients [32,33], further supporting mycophenolate’s unique impact on immunogenicity, regardless of disease background. These observations may have a practical role in clinical management, suggesting that a reduction of the daily dosage may be sufficient to achieve seroconversion in patients with active or early autoimmune diseases, where treatment withdrawal may not be feasible. Our findings provide evidence that reducing mycophenolate dosage to 1 g or less daily may increase vaccine immunogenicity, which may have practical consequences for the management of patients taking this drug and needs to be confirmed by appropriate and rigorous trials. Further, based on our data, we cannot infer information about the optimal timing for drug reduction. As the updated ACR recommendations suggest a 7 day withdrawal, and a subsequent study [41] confirmed the effectiveness of this strategy, we can only speculate that a similar time period could also be adopted for dose reduction. Further studies are needed to confirm that in more at-risk patients, mycophenolate can be reduced to ≤ 1 g/day for vaccination rather than being discontinued, and to establish the adequate period of dose reduction; further, evaluating the long-term impact of reducing or withdrawing mycophenolate for vaccination would be worthwhile for identifying the best balance between immune response to vaccine and risk of disease reactivation, despite the preliminary 1 month data appearing reassuring [41].

We are aware that our study has strengths and limitations. Among the strengths, this study included well-defined patients with a post-vaccination clinical follow-up and the use of the same vaccine schedule, with the same ACR guidance being applied. Moreover, data on the exclusive use of the Moderna vaccine in patients with autoimmune and chronic inflammatory diseases are indeed limited compared to other mRNA vaccines. Further, our study is the first to propose a dose effect of mycophenolate mofetil. Among the limitations, however, we firstly recognize that in our patient cohort, female subjects were predominant and their age was significantly higher than that in the control cohort. The latter could have an impact on immunogenicity; however, among the patient group, mycophenolate was associated with a significantly worse impact on seroconversion and antibody titers compared to other immunosuppressants besides the controls, which obviates the question of whether these differences are age-related. Second, the limited data on the effects of glucocorticoids at a dosage higher than 10 mg of the prednisone-equivalent is due to the low number of these patients in our cohort. Third, we cannot comment on the effect of methotrexate and JAK inhibitors due to the fact that the patients discontinued these medications as recommended [22]. Fourth, measuring the neutralizing activity and cytotoxic assays was feasible in a limited cohort of patients, and ours may be considered pilot results needing to be confirmed on larger populations. Last, the control population only partially matched the patient population, but it had the strength of including only subjects sharing the same household as the patients.

## 5. Conclusions

In conclusion, we report that the Moderna-1273 mRNA vaccination is effective and safe in patients with rheumatic and autoimmune liver diseases, and that mycophenolate at a daily dosage of ≤ 1 g/day may not significantly affect vaccine immunogenicity.

## Figures and Tables

**Figure 1 vaccines-10-00801-f001:**
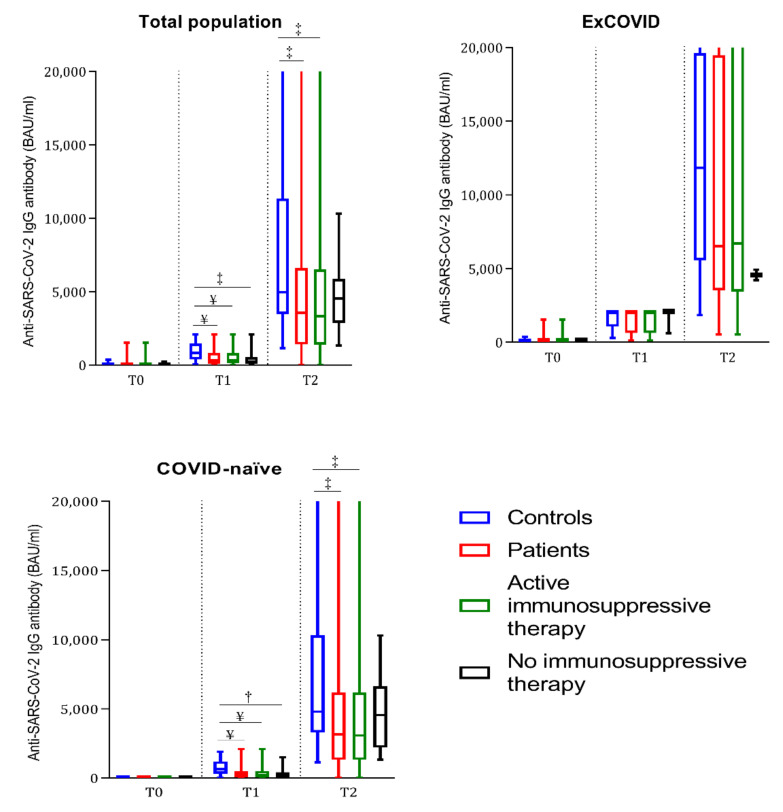
Boxplots representing anti-SARS-CoV-2 IgG antibody titers prior to the first (T0) and second (T1) vaccine dose and after 4 weeks (T2) in controls, total patients, patients under active immunosuppressive therapy, and patients without immunosuppressive therapy. The total population (ex-COVID plus COVID-naïve) and ex-COVID and COVID-naïve subjects are represented in each graph, respectively. † = *p* < 0.01, ‡ = *p* < 0.001, and ¥ = *p* < 0.0001 compared with the COVID-matched controls at the same time point by Mann–Whitney test.

**Figure 2 vaccines-10-00801-f002:**
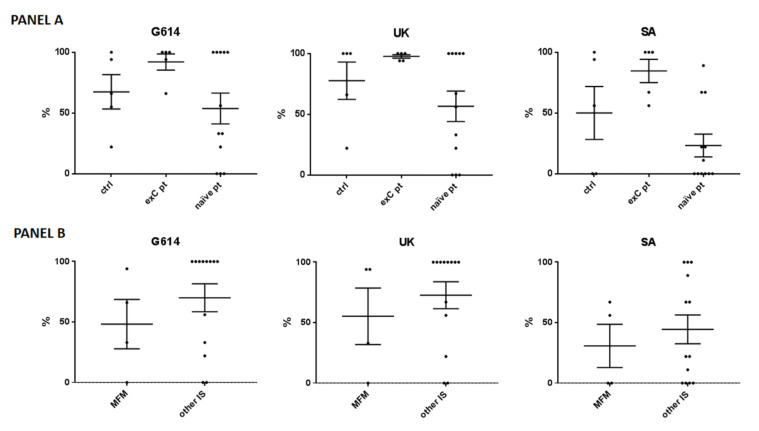
Neutralizing activity against the SARS-CoV-2 wild-type (G614), United Kingdom (UK), and South African (SA) variants. (**Panel A**) The analysis included 5 controls (-ctrl- 3 with previous COVID-19 –exC-), 5 ex-C patients (of which 2 were treated with mycophenolate mofetil (MFM) at 3 g/day, 2 with baricitinib at 4 mg/day, and 1 with methotrexate at 15 mg/week), and 12 COVID-naïve patients (naïve pt), of which 2 treated with MFM at 3 g/day, 2 with baricitinib at 4 mg/day, 2 with methotrexate at 15 mg/week, 2 with secukinumab at 150 mg/4 weeks, and 4 with adalimumab at 40 mg/2 weeks. (**Panel B**) Data are shown for the patients treated with MFM versus other immunosuppressants (other IS). Each black spot represents a patient, the bars indicate means and standard errors of the mean (SEM) of the percentages of cytopathic effect compared to the positive infection control.

**Figure 3 vaccines-10-00801-f003:**
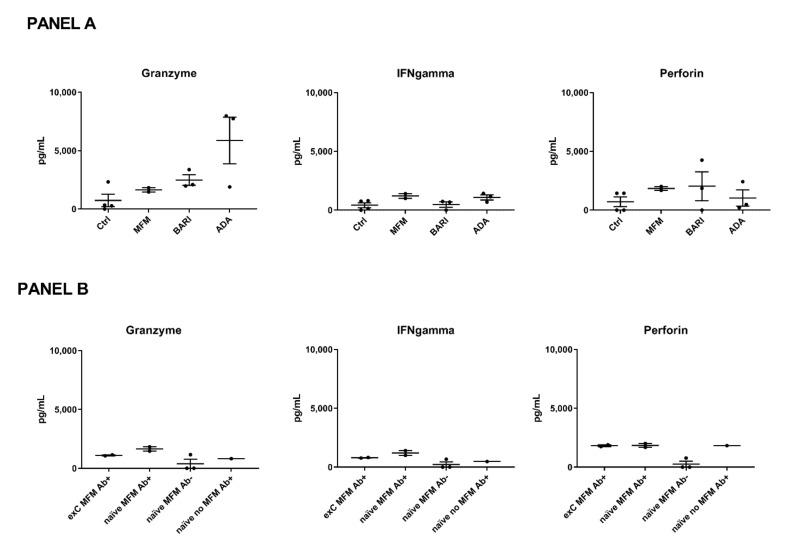
Cytotoxic marker levels. (**Panel A**) Patients included in this analysis were treated with mycophenolate mofetil (MFM, *n* = 2) at 3 g/day, baricitinib (BARI, *n* = 3) at 4 mg/day, and adalimumab (ADA, *n* = 3) at 40 mg/2 weeks. The patients and controls (ctrl, *n* = 4) were all COVID-naïve. (**Panel B**) Patients (*n* = 8) were compared based on MFM treatment, seroconversion status (Ab+ = seroconverted, Ab- = not seroconverted), and previous history of COVID-19 (previous COVID-19= ex-C or COVID-naïve=-naïve). Each black spot represents a patient, the bars indicate means and standard errors of the mean (SEM) of the marker levels.

**Table 1 vaccines-10-00801-t001:** Demographic and clinical features of the study population at the time of enrollment (T0). Treatments with specific medications are illustrated and subdivided into monotherapy (mono) or combined therapy (combo).

	Patients with Immune-Mediated and Chronic Inflammatory Diseases (*n* = 287)	Controls (Caretakers and Family Members; *n* = 67)
Females	175 (61%) †	23 (34%)
Caucasians	277 (96.5%)	66 (98.5%)
Age (years)	55 (19–78) ‡	48 (19–70)
Hypertension	39 (14%) *	2 (3%)
Diabetes	12 (5%)	0
Cardiovascular disease	9 (3%)	0
Ex-COVID	45 (16%)	15 (22%)
Ongoing treatments		
*Mycophenolate mofetil*	28 (10%) monotherapy 7%	
*Glucocorticoids*	43 (15%) monotherapy 1%	-
*Methotrexate*	60 (21%) monotherapy 6%	-
*Azathioprine*	27 (9%) monotherapy 6%	
*AntiTNF-alpha*	108 (38%) monotherapy 30%	-
*Anti-IL17*	20 (7%) monotherapy 5%	-
*Anti-IL6R*	12 (4%) monotherapy 2%	-
*JAK inhibitors*	27 (9%) monotherapy 6%	-
*Other therapies*	21 (7%)	-
*No immunosuppressants*	18 (6%)	67 (100%)
Disease		
*Rheumatoid arthritis*	72 (25%)	-
*Spondyloarthritis*	125 (44%)	-
*Systemic sclerosis*	13 (5%) 12 (92%) on mycophenolate	-
*Systemic lupus erythematosus*	12 (4%) 3 (25%) on mycophenolate	-
*Dermato/Polymyositis*	3 (1%) 3 (100%) on mycophenolate	
*Autoimmune hepatitis*	31 (11%) 7 (23%) on mycophenolate	-
*Primary biliary cholangitis*	13 (5%)	-
*Other rheumatic diseases*	22 (8%) 3 (14%) on mycophenolate	-
Disease duration (months)	60 (2–500)	-
Active disease	81 (28%)	-

Continuous variables are expressed as medians (interquartile ranges); dichotomous variables are expressed as numbers (%). * = *p* < 0.05, † = *p* < 0.01, and ‡ = *p* < 0.0001 compared with controls.

**Table 2 vaccines-10-00801-t002:** Anti-SARS-CoV-2 IgG antibody seroconversion rates and titers prior to the first (T0) and second (T1) vaccine dose and after 4 weeks (T2) in the patients and controls based on previous COVID-19 history.

	T0	T1	T2
**Controls**			
*Overall* titer	4 (4–22.9)	835 (384.5–1455)	4960 (3465–11350)
seroconverted	14/67 (20.9%)	64/65 (98.4%)	45/45 (100%)
*ex-COVID* titer	116 (44.8–238)	2081 (1060–2081)	11850 (5563–19625)
seroconverted	14/15 (93.3%)	15/15 (100%)	8/8 (100%)
*COVID-naïve* titer	4 (4–4)	632.5 (319.8–1195)	4790 (3290–10325) *^p^* ^< 0.0001 vs. MFM^
seroconverted	-	49/50 (98%)	37/37 (100%)
**Patients**			
*Overall* titer	4 (4–4)	284 (91.8–746) *^p^* ^< 0.0001^	3505 (1398–6520) *^p^* ^= 0.0002^
seroconverted	38/287 (13.2%)	239/281 (85.1%) *^p^* ^= 0−006^	211/219 (96.3%)
*ex-COVID* titer	99.1 (58.7–271)	2081 (634.5–2081)	6520 (3518–19475)
seroconverted	38/45 (84.4%)	45/45 (100%)	36/36 (100%)
*COVID-naïve* titer	4 (4–4)	222 (68.3–489) *^p^* ^< 0.0001^	3160 (1328–6170) *^p^* ^= 0.0002^
seroconverted	-	194/236 (82.2%) *^p^* ^= 0.009^	175/183 (95.6%)
**Active immunosuppressive therapy**			
*Overall* titer	4 (4–4)	293.5 (94.4–779.5) *^p^* ^< 0.0001^	3280 (1390–6505) *^p^* ^= 0.002^
seroconverted	35/268 (13%)	223/262 (85.1%) *^p^* ^= 0.006^	197/206 (95.6%)
*ex-COVID* titer	98.2 (51.4–282.3)	2081 (635.3–2081)	6700 (3423–21400)
seroconverted	35/42 (83.3%)	42/42 (100%)	34/34 (100%)
*COVID-naïve* titer	4 (4–4)	225.5 (67–518) *^p^* ^< 0.0001^	3035 (1305–6128) *^p^* ^= 0.006^
seroconverted	-	181/221 (81.9%) *^p^* ^= 0.008^	165/172 (95.9%)
**No immunosuppressive therapy**			
*Overall* titer	4 (4–6.4)	215 (80.6–541) *^p^* ^= 0.0002^	4540 (2875–5865)
seroconverted	3/18 (16.6%)	15/18 (83.3%) *^p^* ^= 0.04^	13/13 (100%)
*ex-COVID* titer	143 (88.5–221)	2081 (622–2081)	4565 (4210–4920)
seroconverted	3/3 (100%)	3/3 (100%)	2/2 (100%)
*COVID-naïve* titer	4 (4–4)	169 (71–409) *^p^* ^= 0.004^	4540 (2200–6630)^*p* = 0.003 vs. MFM^
seroconverted	-	12/15 (80%)	11/11 (100%)
**Active disease**			
*Overall* titer	4 (4–4)	222.5 (80.5–498.8) *^p^* ^< 0.0001^	3220 (1760–7085) *^p^* ^= 0.002^
seroconverted	9/81 (11.1)	64/76 (84.2) *^p^* ^= 0.009^	55/58 (94.8)
*ex-COVID* titer	99.1 (65.5–225)	2081 (502–2081)	5070 (3220–7350)
seroconverted	9/9 (100)	9/9 (100)	7/7 (100)
*COVID-naïve* titer	4 (4–4)	186 (65–332) *^p^* ^< 0.0001^	3040 (1655–7118) *^p^* ^= 0.002^
seroconverted	-	55/67 (82.1) *^p^* ^= 0.01^	48/51 (94.1)
**Inactive disease**			
*Overall* titer	4 (4–5.16)	343 (95.2–827) *^p^* ^< 0.0001^	3600 (1370–6435) *^p^* ^= 0.0004^
seroconverted	29/206 (14)	175/205 (85.3) *^p^* ^= 0.008^	156/161 (96.8)
*ex-COVID* titer	98.7 (46–272.5)	2081 (638–2081)	6820 (3545–21900)
seroconverted	32/36 (88.8)	36/36 (100)	36/36 (100)
*COVID-naïve* titer	4 (4–4)	239 (69.4–608.5) *^p^* ^< 0.0001^	3210 (1303–5660) *^p^* ^= 0.002^
seroconverted	-	139/169 (82.2) *^p^* ^= 0.01^	127/132 (96.2)

Titers are expressed as medians (interquartile ranges) (BAU/mL). The prevalence of seroconversion is expressed in ratios (%). Significant *p* values (*p*) are detailed in the table and indicate a comparison with COVID-matched controls at the same time point; when specified, they indicate comparison with COVID-matched patients under mycophenolate therapy (vs. MFM).

**Table 3 vaccines-10-00801-t003:** Anti-SARS-CoV-2 seroconversion rates and titers in COVID-naïve patients prior to the second (T1) vaccine dose and after 4 weeks (T2) in COVID-naïve patients based on diagnoses and therapies.

	T1	T2
**Rheumatoid arthritis** titer	155 (50–484) ^*p* < 0.0001^	2310 (1230–6375) *^p^* ^= 0.007^
seroconverted	49/61 (80.3%) *^p^* ^= 0.01^	49/49 (100%)
**Spondyloarthritis** titer	317 (133–622) *^p^* ^= 0.02^	4040 (2230–6385)
seroconverted	103/108 (95.3%)	79/79 (100%)
**Systemic sclerosis** titer	4 (4–4) *^p^* ^< 0.0001^	25.6 (8.1–380) *^p^* ^< 0.0001^
seroconverted	2/11 (18.1%) *^p^* ^< 0.0001^	3/11 (27.2%) *^p^* ^< 0.0001^
**Systemic lupus erythematosus** titer	77.5 (5.6–464) *^p^* ^= 0.005^	1320 (498–3100) *^p^* ^= 0.007^
seroconverted	7/11 (58.3%) *^p^* ^= 0.002^	10/10 (100%)
**Dermatomyositis** titer	4 (4–4.87) *^p^* ^= 0.001^	134 (121–178) *^p^* ^= 0.004^
seroconverted	0/3 (0%) *^p^* ^< 0.0001^	3/3 (100%)
**Autoimmune hepatitis** titer	323 (75–556) *^p^* ^= 0.04^	3280 (1300–5340)
seroconverted	23/26 (88.4%)	19/19 (100%)
**Mycophenolate mofetil**		
*overall* titer	4 (4–43.8) *^p^* ^< 0.0001^	156 (27–353) *^p^* ^< 0.0001^
seroconverted	6/23 (26%) *^p^* ^< 0.0001^	14/22 (63.6%) *^p^* ^= 0.0004^
*monotherapy* titer	4 (4–7.12) *^p^* ^< 0.0001^	121 (13.8–517) *^p^* ^< 0.0001^
seroconverted	3/17 (17.6%) *^p^* ^< 0.0001^	9/16 (56.2%) *^p^* ^= 0.0001^
*with glucocorticoids* titer	24.3 (4–81.8) *^p^* ^= 0.001^	399 (141–1948)
seroconverted	3/6 (50%)	5/6 (83.3%)
**Glucocorticoids**		
*monotherapy* titer	229 (222–235)	5270 (-)
seroconverted	3/3 (100%)	1/1 (100%)
*with other therapies* titer	85 (7.6–222) *^p^* ^< 0.0001^	1620 (797–6650) *^p^* ^= 0.005^
seroconverted	22/35 (62.8%) *^p^* ^< 0.0001^	26/27 (96.3%)
**Methotrexate**		
*monotherapy* titer	285 (104–663) *^p^* ^= 0.03 vs. MFM^	3160 (2470–6690)^*p* = 0.003 vs. MFM^
seroconverted	13/14 (92.8%)	11/11 (100%)
*with glucocorticoids* titer	87 (40–638)	9220 (8500–9640)
seroconverted	3/4 (75%)	4/4 (100%)
**Azathioprine**		
*monotherapy* titer	492 (240–825) *^p^* ^< 0.0001 vs. MFM^	3280 (1735–5960) *^p^* ^= 0.02 vs. MFM^
seroconverted	16/16 (100%)	13/13 (100%)
*with glucocorticoids* titer	37.8 (20.9–461^)^ *^p^* ^= 0.003^^, *p* = 0.03 vs. AZA mono^	1620 (804–6605)
seroconverted	4/7 (57.1%)	5/5 (100%)
**Anti-TNFalpha**		
*monotherapy* titer	382 (177–790) *^p^* ^< 0.0001 vs. MFM^	3405 (1715–5340) *^p^* ^= 0.0002 vs. MFM^
seroconverted	66/69 (95.6%)	54/54 (100%)
*with glucocorticoid* titer	67.8 (4–326)	1370 (1320–6659)
seroconverted	2/3 (66.6%)	3/3 (100%)
*with methotrexate* titer	231 (89.3–899)	4130 (1830–6340)
seroconverted	11/13 (84.6%)	11/11 (100%)
**Anti-IL17**		
*monotherapy* titer	228 (96.9–332) *^p^* ^= 0.02^	3105 (2485–4850)
seroconverted	15/15 (100%)	10/10 (100%)
**Anti-IL6R**		
*monotherapy* titer	262 (109–1043)	3700 (2020–5290)
seroconverted	6/6 (100%)	5/5 (100%)
**JAK inhibitors**		
*monotherapy* titer	140 (51–493) *^p^* ^= 0.04^	3750 (1039–6775)
seroconverted	13/15 (86.6%)	10/10 (100%)
*with methotrexate* titer	33.9 (8.2–262.2) *^p^* ^= 0.02^	512 (-)
seroconverted	2/4 (50%) *^p^* ^= 0.004^	1/1 (100%)

Titers are expressed as medians (interquartile ranges) (BAU/mL). The prevalence of seroconversion is expressed in ratios (%). Significant *p* values (*p*) are detailed in the table and indicate a comparison with the COVID-matched controls at the same time point; when specified, they indicate a comparison with the COVID-matched patients under mycophenolate therapy (vs. MFM) or azathioprine (vs. AZA).

**Table 4 vaccines-10-00801-t004:** Demographic and clinical features of the patients taking mycophenolate >1 g daily.

	Patients on MFM >1 g/daily (*n* = 20)
Females Caucasians	18 (90%) 18 (90%)
Age (years)	59.5 (48.5–64.5)
Hypertension	1 (5%)
Diabetes	0
Cardiovascular disease	0
Lymphocyte count at baseline (*10^3^/mm^3^)§ Disease	1700 (960–2100)
*Systemic sclerosis*	11 (55%)
*Systemic lupus erythematosus*	3 (15%)
*Dermato/Polymyositis*	2 (10%)
*Autoimmune hepatitis*	2 (10%)
*Primary biliary cholangitis*	1 (5%)
*Other rheumatic diseases*	1 (5%)
Disease duration (months)	62 (27–132)
Active disease Seroconversion at T2 in COVID-naïve patients	3 (15%) 8 (40%)
Antibody titer at T2 (BAU/mL) in COVID-naïve patients	34.2 (9.8–443.5)

Continuous variables are expressed as medians (interquartile ranges); dichotomous variables are expressed as numbers (%). The § total lymphocyte count at baseline was considered as a marker of baseline immune state and was available for 15/20 patients (75%).

**Table 5 vaccines-10-00801-t005:** Cumulative adverse events observed post-vaccination according to the patients’ and controls’ previous COVID-19 history.

Patients (*n* = 219)	
***Adverse events*** Mild Moderate Severe	152/219 (69.4%) 105/219 (48%) 46/219 (21%) 1/219 (0.4%)
Adverse events in ex-COVID	22/30 (73.3%)
Adverse events in COVID-naïve	127/184 (69%)
* **Anti-SARS-CoV-2 titer** *	
COVID-naïve with adverse events COVID-naïve without adverse events	3265 (1375–6423) 2570 (1170–4973)
**Controls** (*n* = 45)	
* **Adverse events** *	35/45 (77.7%)
Mild Moderate Severe	26/45 (57.7%) 9/45 (20%) 0/45 (0%)
Adverse events in ex-COVID	7/8 (87.5%)
Adverse events in COVID-naïve	28/38 (73.6)
* **Anti-SARS-CoV-2titer** *	
COVID-naïve with adverse events COVID-naïve without adverse events	6525 (3950–11,375) * 3300 (2770–3855)

Mild adverse events included pain at the injection site, nausea, low-grade fever (<38 °C), fatigue, or headache, and they did not interfere with daily and work activities and lasted less than 2 days. Moderate adverse events included fever (>38 °C), vomiting, or herpes zoster reactivation, and they did interfere with daily and work activities. The severe adverse event occurred in a patient who experienced high blood pressure and metrorrhagia and was evaluated at the emergency department 7 days after the second dose, where treatment for newly diagnosed hypertension was started. The titers are expressed as medians (interquartile ranges) (BAU/mL). The prevalence is expressed as ratios (%). * *p* < 0.001 compared to anti-SARS-CoV-2 titers in COVID-naïve controls without side effects.

## Data Availability

Data are available after reasonable request to the corresponding author.

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
