# Peer review of "Dose-Dependent Impairment of the Immune Response to the Moderna-1273 mRNA Vaccine by Mycophenolate Mofetil in Patients with Rheumatic and Autoimmune Liver Diseases"

_vaccines, 2022, doi:10.3390/vaccines10050801_

Round 1
Reviewer 1 Report
Estimated Authors,
Estimated Editors,
First of all thank you for the opportunity to review this very interesting paper about the vaccination on patients that are recipient of therapy with Mycophenolate mofetil because of underlying rheumatic and autoimmune liver diseases. Mycophenolate mofetil (MMF) is a prodrug of mycophenolic acid (MPA), an inhibitor of inosine monophosphate dehydrogenase (IMPDH). This is the rate-limiting enzyme in de novo synthesis of guanosine nucleotides, a pathway that affects T- and B-lymphocytes eventually impairing the immune system. In fact, the results seemly suggest that MPA/MMF eventually impair an appropriate and long-lasting seroconversion, representing a potential issue in achieving appropriate vaccination rates in some high-risk groups.
Authors have reported about their experience with vaccination by means of Moderna-1273 mRNA vaccine in around 1000 patients, 287 of them being eventually included in the sample. This specific dropout rate represents, from my point of view, the most significant shortcoming of the study, and should be discussed by Authors more extensively. Similarly, Authors should discuss in further details how the heterogenous characteristics of the treatment (see Table 1) may have influenced the overall results of the study.
Another shortcoming of the present study is unfortunately represented by the data reporting. While the text is clear and straightforward in its formulation, at least from the Point of View of the present reviewer, Table 2-3 and Figure 1 require some improvements, both minor and major.
Regarding Figure 1, only minor suggestions: please revise the color scheme in order to improve the readability by avoiding light grey.
More extensive improvements are required by Table 2-3, that must implement specific p values of the comparisons you've performed, avoiding categorization of p value, as to this point reported.
I've no further requests for a paper that is, from my point of view, very well written.
Author Response
Reviewer 1.
Estimated Authors, Estimated Editors,
First of all thank you for the opportunity to review this very interesting paper about the vaccination on patients that are recipient of therapy with Mycophenolate mofetil because of underlying rheumatic and autoimmune liver diseases. Mycophenolate mofetil (MMF) is a prodrug of mycophenolic acid (MPA), an inhibitor of inosine monophosphate dehydrogenase (IMPDH). This is the rate-limiting enzyme in de novo synthesis of guanosine nucleotides, a pathway that affects T- and B- lymphocytes eventually impairing the immune system. In fact, the results seemly suggest that MPA/MMF eventually impair an appropriate and long-lasting seroconversion, representing a potential issue in achieving appropriate vaccination rates in some high-risk groups.
We are thankful to the Reviewer for the encouraging comments and the thorough consideration given to our work.
Authors have reported about their experience with vaccination by means of Moderna-1273 mRNA vaccine in around 1000 patients, 287 of them being eventually included in the sample. This specific dropout rate represents, from my point of view, the most significant shortcoming of the study, and should be discussed by Authors more extensively.
The Reviewer is right and the fact that 287 subjects agreed to participate out of the 1073 patients followed at our Hospital directly solicited by phone requires further clarification. We believe that this discrepancy is related to two major factors. First, at the time of the study, subjects were solicited from different sources to get vaccinated with high priority, including older age or professional risks. Second, more than one third of the patients followed at our Center come from a different region and thus were more likely to prefer to get vaccinated in there are. We detailed this in new page 3, line 113-119.
Similarly, Authors should discuss in further details how the heterogenous characteristics of the treatment (see Table 1) may have influenced the overall results of the study.
Here again the Reviewer is quite correct that the heterogeneous treatments, with different mechanisms of action and impact on the immune system, certainly have influenced the overall results, with antibody titers being significantly lower in patients under immunosuppressive therapy compared to controls. We further analyzed the impact of each treatment as detailed in Table 3, with mycophenolate being the drug with the worst impact on the immune response, both in terms of seroconversion and antibody titer levels. On the other hand, the other immunosuppressants, both for their mechanisms of action and by adopting ACR guidance on short-term withdrawal for vaccination, had a mild, if any, impact on immunogenicity. We have now added more background information on the impact of mycophenolate and other immunosuppressants on response to vaccinations in the Introduction and discussed this further in the Discussion section. Please see new pages 2-3 lines 47-104 and page 16 lines 420-429.
Another shortcoming of the present study is unfortunately represented by the data reporting. While the text is clear and straightforward in its formulation, at least from the Point of View of the present reviewer, Table 2-3 and Figure 1 require some improvements, both minor and major. Regarding Figure 1, only minor suggestions: please revise the color scheme in order to improve the readability by avoiding light grey. More extensive improvements are required by Table 2-3, that must implement specific p values of the comparisons you've performed, avoiding categorization of p value, as to this point reported.
Nice observations indeed. We have now made the suggested changes, particularly in the new Figure 1 and added specific p-values in Tables 2 and 3. We are confident clarity benefits from the editing.
I've no further requests for a paper that is, from my point of view, very well written.
Reviewer 2 Report
The manuscript entitled ''Dose-dependent impairment of the immune response to Moderna-1273 mRNA vaccine by mycophenolate mofetil in patients with rheumatic and autoimmune liver diseases" reported that the Moderna-1273 mRNA vaccination is effective and safe in patients with rheumatic and autoimmune liver diseases and that mycophenolate at a daily dosage ≤1 g/day may not significantly affect vaccine immunogenicity.
The topic is of great interest taking into consideration what are living now. The manuscript deserves to be published as its topic fits with the aim and the topic of the journal even if I would appreciated to see more background and include more relevant references that the authors missed.
Author Response
Reviewer 2.
The manuscript entitled ''Dose-dependent impairment of the immune response to Moderna-1273 mRNA vaccine by mycophenolate mofetil in patients with rheumatic and autoimmune liver diseases" reported that the Moderna-1273 mRNA vaccination is effective and safe in patients with rheumatic and autoimmune liver diseases and that mycophenolate at a daily dosage ≤1 g/day may not significantly affect vaccine immunogenicity.
The topic is of great interest taking into consideration what are living now. The manuscript deserves to be published as its topic fits with the aim and the topic of the journal even if I would appreciated to see more background and include more relevant references that the authors missed.
We are thankful for the very encouraging comments to our work. We have implemented the introduction with more background information on the current knowledge of the impact of mycophenolate and other immunosuppressants on the response to vaccinations and have added new references. Please see the new pages 2-3 lines 47-104.
We also run a spellcheck and believe to have improved the English language.
We hope our manuscript is now more complete and acceptable for publication in Vaccines.
Reviewer 3 Report
This stuy provides rather convincing evidence that Mycophenolate treatment may affect response to the Moderna mRNA vaccine with a dose dependent effect. Thus, reducing the daily dosage to less than 1g may increase vaccine immunogeniciity. I feel that the findings of this exploratory study may have practical consequences if they are confirmed by appropriate controlled trials. This hsould be better explained in the text. The authors should also comment, among the limits of the study, about possible selective drop-out from T1 to T3. Could the lost of fpollow-up have affected the reults of the study? Moreover, the study population was recruited among a larger number of patients with rheumatic or autoimmune diseases. What about the possibility of a selection bias?
Minor points (abstract): are 95% CI instead of reporting also sensitivity and specificity on a limited number of participants? In the last sentence, instead of "which is not influenced" I would rephrased as "independently on rheumatological disease.
Author Response
Reviewer 3.
This study provides rather convincing evidence that Mycophenolate treatment may affect response to the Moderna mRNA vaccine with a dose dependent effect. Thus, reducing the daily dosage to less than 1g may increase vaccine immunogeniciity. I feel that the findings of this exploratory study may have practical consequences if they are confirmed by appropriate controlled trials. This should be better explained in the text.
We agree and it is frustrating when we cannot make our points clear like we should. Nonetheless, we agree with the Reviewer that our data support a dose-dependent impact of mycophenolate on immune response to Moderna mRNA vaccine, evidence that needs to be confirmed by rigorous trials. We underlined this point in the discussion; please see new page 16 lines 463-466.
The authors should also comment, among the limits of the study, about possible selective drop-out from T1 to T3. Could the lost of follow-up have affected the results of the study?
Nice catch, dropouts are often overlooked in real-world studies and indeed we cannot speculate on ‘what if’ all patients showed up and were investigated at all timepoints. On the day of the first vaccine dose (T0) 287 patients and 67 controls were enrolled and on the day of the second vaccine dose 281 (98%) patients and 65 (97%) controls have continued the study, which became 219 (76%) patients and 45 (67%) controls at T2 (4 weeks after the second dose), mainly because some subjects felt uncomfortable to undergo a new blood withdrawal. To evaluate the late impact on immune response, we added a further timepoint 8 weeks after the second dose (T3), where we analyzed only the patients, in particular those under mycophenolate (10 patients agreed to continue the study) compared with other 37 patients (22 treated with anti-TNF-alpha, 6 with JAK inhibitors, and 9 not on immunosuppressive treatment). This latter population was randomly selected and constituted a representative sample of the initial population. We detailed this is new page 3, lines 135-142.
Moreover, the study population was recruited among a larger number of patients with rheumatic or autoimmune diseases. What about the possibility of a selection bias?
This concern is shared also by reviewer #1 and of course both are right. As replied in a previous point, the fact that 287 subjects agreed to participate out of the 1073 patients followed at our Hospital directly solicited by phone requires some explanation. We believe that this discrepancy is related primarily to the fact that, at the time of the study, Italian citizens with rheumatic diseases were likely to be solicited for vaccination by different agencies / health care providers, because of older age or professional risks. Alternatively, more than one third of the patients followed at our Center come from a different region and thus were more likely to prefer to get vaccinated in there. We detailed this is new page 3, line 113-119.
Minor points (abstract): are 95% CI instead of reporting also sensitivity and specificity on a limited number of participants? In the last sentence, instead of "which is not influenced" I would rephrased as "independently on rheumatological disease.
We thank the Reviewer for the thorough revision and made the suggested changes, we also checked and improved English language and we hope our manuscript is now acceptable.